# Associations between Sleep Quality, Frailty, and Quality of Life among Older Adults in Community and Nursing Home Settings

**DOI:** 10.3390/ijerph20064937

**Published:** 2023-03-10

**Authors:** Mateja Lorber, Sergej Kmetec, Adam Davey, Nataša Mlinar Reljić, Zvonka Fekonja, Barbara Kegl

**Affiliations:** 1Faculty of Health Sciences, University of Maribor, Zitna Ulica 15, 2000 Maribor, Slovenia; 2College of Health Sciences, University of Delaware, 210 South College Avenue, Newark, NJ 19716, USA

**Keywords:** sleep quality, frailty, quality of life, older adults, nursing homes, community-dwelling

## Abstract

Poor sleep quality is prevalent among older adults, but limited data document associations between frailty and quality of life comparing individuals living in the community with those in nursing homes. This cross-sectional study (conducted between August and November 2019) included 831 older adults (mean age 76.5 years) from Slovenia’s community and nursing home settings. The results showed comorbidity in 38% of community-dwelling older adults and 31% of older adults in nursing homes. The prevalence of frailty among community-dwelling older adults was 36.5%, and among older adults in a nursing home was 58.5%. A total of 76% of community-dwelling older adults and 95.8% of nursing home residents reported poor sleep quality. Sleep quality and frailty predict 42.3% of the total variability of quality of life for older adults in nursing homes and 34.8% for community-dwelling older adults. The study’s results indicate that the quality of life can be affected by factors (e.g., worse sleep quality and frailty) among older adults, regardless of being a resident or from the community. Understanding how sleep quality is affected by social, environmental, and biological factors can help improve sleep quality and potentially the quality of life of older adults.

## 1. Introduction

Frailty is a common old-age syndrome with significant implications for public health [1] and has emerged as a major public health concern [2] and is important when considering financial healthcare planning [3]. Frailty represents a loss of physiological reserve and resistance to stressors due to cumulative declines across multiple physiological systems [4,5]. As a result, it makes older people more vulnerable to poor health outcomes [2]. Increasing life expectancy brings opportunities for older adults, their families, and societies [6], but a greater emphasis should be placed on identifying risk factors to avoid or reduce frailty in older adults—especially those with one or more chronic diseases [7]. The prevalence of frailty is much higher among older adults living in nursing homes than in community settings, and it is a significant predictor of mortality among older adults in nursing homes.

Frailty in older adults is associated with sleep quality. Poor sleep quality is prevalent among older adults and is compounded by frailty, significantly correlating with poor sleep quality in older adults living in nursing homes [8,9]. Algahtani 2021 demonstrated a significant association between frailty and poor sleep quality in older adults living in a community-dwelling sample with chronic diseases. In addition to the physiological changes of aging, sleep quality in old age is also affected by chronic diseases, which are increasingly prevalent with age [10]. The results in 15 European countries between 2004 and 2017 showed considerable variability in the prevalence of comorbidity in adults aged 50. The prevalence of comorbidity increases between the first and second waves of research. In Slovenia in 2017, the rate of comorbidity was about 40%; the lowest percentage (approximately 30%) was in The Netherlands, Switzerland, and Sweden. In all other European countries, the rate was higher than 40%, up to 55%. In all countries, the percentage of comorbidity increases with age [11]. The rate of comorbidity is much higher in the United States (62.5%) [12]. A growing body of research indicates that insufficient sleep may also increase the risk of several conditions and chronic diseases, including diabetes, cardiovascular disease, obesity, and depression [13]. Decreasing sleep quality can result from several factors, such as physiological changes in age, underlying physical conditions, and psychosocial factors [14]. Poor sleep quality can harm older adults’ physical and mental health and daytime functioning [15,16]. Some studies indicate more than half of older adults have sleep problems [17], and 5–8% of older adults experience insomnia [18]. Sleep disorders indicate the possibility of disturbances in circadian rhythms in chronic diseases, which significantly worsen the quality of life [19]. In an aging population, multimorbidity has been identified as a big challenge for patients and health systems worldwide [20].

Andersen’s behavioral model [21] was used for factors associated with older adults’ quality of life. The model includes contextual and individual characteristics, including demographics, social, human, and material resources, and factors such as several chronic conditions. Health behaviors include physical activity and support. The decline in one or more of a person’s abilities is associated with reductions in quality of life [22].

According to the literature review, only the association between sleep quality and quality of life or frailty and quality of life has been investigated so far, and this research included only community-dwelling elderly. Because until now there has been no research that included all three variables, we investigated the association between sleep quality, frailty, and quality of life among community-dwelling older adults and nursing home residents.

## 2. Materials and Methods

### 2.1. Study Design

This study is based on a cross-sectional approach by using a questionnaire. We followed the STROBE checklist for reporting cross-sectional studies [23] for rigorous reporting of the study results.

### 2.2. Setting and Participants

This study used convenience sampling to invite older adults into nursing homes and community dwellings in the northeastern part of Slovenia. The inclusion criteria for the sample were older adults (65+ years) living in nursing homes or community dwellings and cognitive functioning appropriate to complete the survey (cognitive ability was assessed so that the healthcare professional considered which participants could participate in the study. In doing so, they looked at the absence of known organic or psychiatric affecting cognitive ability). There were no exclusion criteria. The sample size was calculated (using the Cochran formula) to allow the detection of clinically meaningful associations, i.e., R^2^ ≥ 0.02 within each group, which suggested *n* = 387 (*e* = 95%; *z* = 5%) in each setting.

### 2.3. Data Collection and Analysis

Participants completed the Pittsburgh Sleep Quality Index (PSQI) [24], the Tilburg Frailty Indicator (TFI) [25], the Quality of life self-assess question on a five-point Likert scale (from 1 “very poor” to 5 “very good”) and 11 demographic and personal questionnaires (demographics, physical activity, chronic disease, income, social interaction, support, etc.).

PSQI is a 19-item questionnaire measuring sleep quality and quantity among adults. This questionnaire consists of seven domains (subjective sleep quality, sleep latency, duration, habitual sleep efficiency, sleep disturbances, use of sleep medication, and daytime dysfunction). Overall scores range from 0 to 21. A higher score indicates poorer sleep quality and a high level of sleep disorders. A global score ≥ 5 indicates clinically significant sleep impairment [24].

TFI is a 15-item questionnaire measuring frailty among older adults. TFI consists of three components (physical, psychological, and social components). Total scores range from 0 to 15 [25], with total scores ≥ 5 points indicating frailty [25,26].

Data were collected from August to November 2019 in Slovenian nursing homes and the local community in the northeastern part of Slovenia based on prior ethical approval and written permission from participating community registered nurses who assisted us with preventive or curative visits to include older adults living in the community. Of the 2000 paper-based questionnaires distributed, 872 were returned. After removing 41 questionnaires because of missing data, we had 831 questionnaires fully completed for further analysis (response rate was 41.5%).

Continuous variables are presented as means (*M*) and standard deviations (*SD*) (normally distributed) or as median (*Mdn*) with interquartile range (IQR) (not normally distributed). Categorical variables were presented as frequencies and percentages. The Shapiro–Wilk test was used to check the normality distribution of variables. Mann–Whitney U tests compared participant groups’ mean scores for continuous or ordinal variables. Correlations between sleep quality, frailty, and quality of life regarding the participant group were checked with Spearman’s correlation coefficient [27]. All analyses use α = 0.05 to determine statistical significance. All three questionnaires showed acceptable internal validity as the Cronbach α coefficient exceeded our desired threshold of 0.70 [28].

### 2.4. Ethical Considerations

Ethical review and approval were obtained (Ref.: 038/2018/2510-1/504). We also obtained permission from the nursing homes and individual participants to conduct the study. Participants in the study were informed in writing of the purpose and objectives, confidentiality, anonymity, and voluntary withdrawal from participation before submitting the questionnaire. The study strictly adheres to the ethical principles of the Declaration of Helsinki [29] and the Oviedo Convention provisions [30].

## 3. Results

The study included 831 participants, mostly women (66.5%; *n* = 553), with a mean age of 76.5 years (SD = 1.6). Similarly, a higher percentage of participants from nursing homes (70%; *n* = 231) and the community (64.3%; *n* = 322) were female. On average, participants from nursing homes were 79.0 years old (SD = 9.0), and the mean age in community dwellings was 74.9 years (SD = 9.5). Modal relationship status was widowed overall (50. 7%; *n* = 421); 71.2% (*n* = 235) of nursing home residents were widowed, whereas 42. 9% (*n* = 215) of community residents were married. Most commonly, participants had completed secondary education overall (54.5%; *n* = 453), as well as in nursing homes (46.1%; *n* = 152) and the community (60.1%; *n* = 301). Overall, 280 participants took medication for sleep disorders. One hundred and fifty participants regularly took medication for insomnia, and 130 participants took it less than once a week. Most participants had one chronic disease overall (52.2%; *n* = 434), in nursing homes (47.9%; *n* = 158), and in the community (55.1%; *n* = 276). Two to three chronic diseases accounted for the second largest percentage regarding the number of chronic diseases (Overall: 31.5%; *n* = 262; Nursing homes: 37%; *n* = 122; Community: 27.9%; *n* = 140). Most participants overall had at least one chronic disease (65.2%; *n* = 542), as did participants in nursing homes (73.6%; *n* = 243) and participants in the community (59.7%; *n* = 299). Overall, 77.5% (*n* = 611) of participants were satisfied with their living environment, as well as in nursing homes (69.4%; *n* = 229) and the community (76.2%; *n* = 382) (Table 1). Among chronic diseases, most had arterial hypertension (*n* = 497), followed by diabetes (*n* = 327), asthma (*n* = 68) and chronic obstructive pulmonary disease (*n* = 6).

As shown in Table 1, sleep quality scores differed between participants in nursing homes (*M* = 10; *SD* = 3.63) and in the community (*M* = 6.84; *SD* = 3.06, U = 42,104.0; *p* < 0.001). Sleep quality scores also differed between men and women in community settings (*U* = 21,255. 5; *p* < 0.001), but not among nursing home residents. Data also indicate a significant association between sleep quality and age among community-dwelling participants (*r_s_* = 0.227; *p* < 0.001). Sleep quality was also associated with relationship status (*χ*^2^(4)_Nursing home_ = 27.123; *p*_Nursing home_ < 0.001; χ^2^(4)_Community_ = 75.123; *p*_Community_ < 0.001). The quality of sleep in community dwellers differs by chronic disease diagnosis (*U* = 12,003.0; *p* < 0.001; *M* = 8.48, *SD* = 3.18). For participants in nursing homes, their sleep differs if they did not have an intimate relationship (*U* = 849.5; *p* < 0.001; *M* = 13.92, *SD* = 4.50) and if they were satisfied with the living environment or not (*U* = 9355.5; *p* < 0.001; *M* = 9.45, *SD* = 3.41).

Data indicate a significant difference in the prevalence of frailty between participants in nursing homes (*M* = 5.56; *SD* = 2.71) and community settings (*M* = 4.32; *SD* = 3.14, *U* = 59,181; *p* < 0.001). There is a difference in the mean scores related to gender between these two groups (*M*_Nursing home_ = 10.04; *SD*_Nursing home_ = 3.63; *M*_Community_ = 4.59; *SD*_Community_ = 3.15), which remains statistically significant for both groups (*U*_Nursing home_ = 8331.0; *p*_Nursing home_ < 0.001; *U*_Community_ = 24,499.5; *p*_Community_= 0.005). There is also a statistical correlation between frailty and age among nursing homes (*r_s_* = 0.310; *p* < 0.001) and community dwellings participants (*r_s_* = 0.250; *p* < 0.001). The level of education and the prevalence of frailty was statistically associated with participants from nursing homes (*χ*^2^(2) = 16.038; *p* = 0.003) and community residences (*χ*^2^(2) = 18.059; *p* < 0.001). There is a statistical association between the number of chronic non-communicable diseases and the prevalence of frailty in the nursing home participants (*χ*^2^(3) = 12.408; *p* = 0.006), but not among community residents. However, there is a statistical association between the type of chronic non-communicable disease and the prevalence of frailty in participants from the community dwellings (*χ*^2^(2) = 9.832; *p* = 0.007). In both groups, there is also a statistically significant association between the diagnosis of severe chronic disease and the prevalence of frailty (*χ*^2^(4)_Nursing home_ = 27.123; *p*_Nursing home_ < 0.001; *χ*^2^(4)_Community_ = 75.123; *p*_Community_ < 0.001).

Quality of life differed significantly (*U* = 64,849.5; *p* < 0.001) between participants from nursing homes (*M* = 3.13; *SD* = 1.06) and those from community dwellings (*M* = 3.51; *SD* = 0.95). Quality of life is higher among men than women among nursing home participants (*U* = 9659.0; *p* = 0.020). In addition, there is also a difference in quality of life in both groups regarding relationship status (*χ*^2^(4)_Nursing home_ = 11.606; *p*_Nursing home_ = 0.021; *χ*^2^(4)_Community_ = 25.534; *p*_Community_ < 0.001). There is also a difference in quality of life regarding education for both groups (*χ*^2^(2)_Nursing home_ = 6.512; *p*_Nursing home_ = 0.039; *χ*^2^(2)_Community_ = 15.534; *p*_Community_ < 0.001). In addition, quality of life is lower among community-dwelling older adults with one or more chronic disease (*U* = 12,964.5; *p* < 0.001; *M* = 3.02, *SD* = 1.01). Among participants from nursing homes, there is a difference in the quality of life if they had experienced a divorce or breakup of an intimate relationship (*U* = 684.5; *p* < 0.001; *M* = 1.75, *SD* = 1.14). Groups differed in quality of life regarding satisfaction/dissatisfaction with their living environment (*U*_Nursing home_ = 6471.5; *p*_Nursing home_ < 0.001; *U*_Community_ = 17,769.0; *p*_Community_ < 0.001).

Table 2 shows the Spearman correlations among participants’ reported sleep quality, frailty, and quality of life. Correlations for participants from nursing homes range from 0.647 to 0.418, and for community-dwelling participants, range from 0.533 to 0.467.

Spearman’s correlation (Table 2) was used to determine associations between the two participant groups’ sleep quality, frailty, and quality of life scores. There was a strong positive correlation between poor sleep quality and frailty in participants from nursing homes (*r_s_* = 0.418, *n* = 330, *p* < 0.001) and a strong positive correlation in participants from the community-dwelling group (*r_s_* = 0.467, *n* = 501, *p* < 0.001). Both groups had a strong negative correlation between frailty and quality of life (*r_s_* _Nursing home_ = −0.435, *n*_Nursing home_ = 330, *p*_Nursing home_ < 0.001; *r_s_* _Community_ = −0.524, *n*_Community_ = 501, *p*_Community_ < 0.001). Both participant groups also had a strong negative correlation between sleep quality and quality of life (*r_s_* _Nursing home_ = −0.647, *n*_Nursing home_ = 330, *p*_Nursing home_ < 0.001; *r_s_*
_Community_ = −0.533, *n*_Community_ = 501, *p*_Community_ < 0.001).

We also looked at whether prediction variables (gender, chronic disease, income, social interaction, support (formal or informal), physical activity, experience in the past year: with the death of a loved one) predict sleep quality, frailty, and quality of life in a nursing home and community-dwelling participants. Overall, regression models for nursing home and community-dwelling participants were statistically significant. Results for sleep quality showed that the predictor variables explained 28.4% of the variance among nursing home participants (F(10,490) = 18.223, *p* < 0.001) and 23.3% of the variance for community-dwelling participants (F(10,490) = 21.340, *p* < 0.001). Models explained 18.6% of the variance for frailty among participants in the nursing homes (F(10,319) = 10.481, *p* < 0.001) and 45.0% for community-dwelling participants (F(10,490) = 57.539, *p* < 0.001). Predictors explain 22.4% of the variance in quality of life for participants in nursing homes (F(10,319) = 13.279, *p* < 0.001) and 34.3% for community-dwelling participants (F(10,490) = 36.776, *p* < 0.001).

In turn, sleep quality and frailty explain 42.3% of the variance in quality of life among older adults in nursing homes (F(2327) = 119.712, *p* < 0.001) and 34.8% for community-dwelling older adults (F(2498) = 132.933, *p* < 0.001).

## 4. Discussion

Our findings indicate that poor sleep quality and frailty are associated with the quality of life among older adults living in community and nursing home settings. This study allows direct comparisons between community-dwelling older adults or residents from nursing homes for all studied variables. Some previous studies indicated that poor sleep quality is significantly related to the frailty of community-dwelling older adults [4,31,32] and older adults in aged care homes [9] and in middle-aged populations [33,34]. In a systematic review of seven studies, Wai and Yu [35] noted evidence for an association between perceived sleep quality and frailty among older adults. Similarly, in their meta-analysis, Pourmotabbed, et al. [35] noted that longer and shorter sleep durations are associated with an increased risk of frailty.

Eighty-five percent of older adults (community-dwelling and nursing home) have one or more chronic diseases. Comorbidity (two or more chronic conditions) is present in 38% of community-dwelling older adults and 31% of older adults in nursing homes and is also important for understanding sleep quality. These results are encouraging because Sheikh, et al. [36] noted that chronic disease is prevalent globally and that more than half of older adults experience comorbidity.

At the same time, 69–76% (69% community-dwelling older adults and 76% older adults in nursing homes) are satisfied with their living environment. These results are important because satisfaction with the living environment is related to quality of life, social satisfaction, and, indirectly, loneliness [37,38]. We also found differences in the quality-of-life score between those who are satisfied or not with their living environment in the community-dwelling older adults and older adults in nursing homes.

Our study found that frailty prevalence was 36.5% among community-dwelling older adults and 58.5% for older adults in nursing homes in our sample, partially comparable with previous results in other countries only for community-dwelling older adults. In China, the most frailty among older adults with chronic diseases in rural areas was 21% [31]. Hladek et al. [39] found that 49% of community-dwelling people with at least one chronic disease in the United States were frail/prefrail.

Determining poor sleep quality by a cut-off global PSQI score ≥ five found that 76.0% of community-dwelling older adults reported poor sleep quality, and 95.8% of older adults from nursing homes reported poor sleep quality. A study from Malaysia reported that approximately 95% of participants from nursing homes reported poor sleep quality [9].

The analysis found that older adults’ relationship status, physical activity, social interaction, support, chronic disease, and income affect the sleep quality, frailty, and quality of life of community-dwelling older adults and residents in nursing homes. With these factors, we can explain 28.4% of the variance of sleep quality for older adults in nursing homes and 23.3% for community-dwelling older adults. With studied factors, we can explain 18.9% of the total variance of the frailty of older adults in nursing homes and 45% for community-dwelling older adults. We can also explain 22.4% of the variance for the quality of life for older adults in nursing homes and 34.3% for community-dwelling older adults. According to our findings for older adults, regardless of whether they live in the community or nursing home, for better sleep quality, prevention of frailness, and higher quality of life, it the most important that they are physically active and have social interaction. In a systematic review, Oliveira, et al. [40] noted that the meta-analysis results showed that physical activity prevents frailty. Wang, Wang, Xie, Liu, and Wang [5] indicated that moderate physical activity is more effective than intensive activity for better sleep quality for older and young populations. Souza, Oliveras-Fabregas, Minobes-Molina, de Camargo Cancela, Galbany-Estragués, and Jerez-Roig [11] noted that included studies indicated positive results connecting physical activity and quality of life of older adults. Moderate evidence suggests that physical activity improves individuals’ quality of life and well-being in all ages [41]. Furthermore, Gothe et al. [42] noted that physical activity and sleep quality were significantly correlated and that sleep quality indirectly influences the quality of life. Tighe et al. [43] pointed out that social contacts, or lack of them, are also important for sleep quality, particularly for older adults, which is in line with our results because older adults are more often prone to loneliness due to life events.

We found that the predicted sleep quality and frailty variables explain 42.3% of the quality-of-life variance for older adults in nursing homes and 34.8% for community-dwelling older adults. According to the findings, monitoring the prevalence of comorbidity and frailty is important for the healthcare system and the factors affected by them. Nevertheless, we must be aware that both comorbidity and frailty are related to disability and dependency in late life worldwide. Moreover, some other studies support our results, for example, that frailty [44,45,46] and poor sleep quality [19,47,48] are related to the quality of life among older adults, even though they are all carried out in the community. Our results show that health-related aspects of quality of life among older adults are related to sleep impairment and confirm that perceived good sleep quality predicts the quality of life of older adults, even if they have no sleep impairment.

The strengths of this study lie in the data complexity and comprehensiveness of data analysis on the association between sleep quality and the incidence of frailty and the resulting impact on the quality of life of older people in nursing homes and community-dwelling. The results are also supported by the fact that no significant differences were detected between the group of older people in the home and the community. The study provides detailed, reliable, and representative cross-sectional findings on differences in sleep quality, frailty, and quality of life between older people in nursing homes and community-dwelling. The extensive survey was subject to rigorous quality control, making it a high-quality data source. Our results show differences in sleep quality, the prevalence of frailty, and the quality of life in older people in nursing homes and the community, which can inform future potential intervention targets.

This study had some limitations that need to be addressed; namely, it used a cross-sectional design, so causal inferences cannot be made. Few studies have been published comparing sleep quality and frailty and their associations with quality of life in older people in nursing homes and the community. Most studies have investigated the connection between each component separately rather than its entirety with all three components. Second, recall bias can occur in a cross-sectional study. Another study limitation could present the type of sampling and its effect on the results. Due to the majority share of one gender and location, we cannot generalize the results to the whole population. In addition, the study used self-assessed data, which could introduce social desirability bias, where respondents over- or under-report their answers based on social norms. The response rate in the study could also be a limitation, as it was 41.5%, which could be considered satisfactory, and no major system failures were found. However, as participation in the study was voluntary, we do not know whether the sample fully represents the key population characteristics. Finally, we could not adjust for all potentially confounding factors nor any potential interaction between sleep quality and paresis (weakness).

## 5. Conclusions

This cross-sectional study is the first to report data on associations between sleep quality, frailty, and quality of life in Slovenians aged 65 and older living in community and nursing homes. We found associations between poorer sleep quality and frailty and poorer quality of life in community-dwelling older people and nursing home residents. A better understanding of how sleep quality is affected by social, environmental, and biological factors may lead to more effective ways to prevent the onset of frailty and its resulting negative impact on the quality of life among older adults. The results of this research can help healthcare professionals formulate strategies to enhance sleep quality among older persons, particularly in nursing home settings where sleep quality is considerably poorer and frailty more prevalent.

## Figures and Tables

**Table 1 ijerph-20-04937-t001:** Demographic variables and their relationship with sleep quality, frailty and quality of life.

Variables		Descriptive Statistics			Sleep Quality (Total)		Frailty (Total)		Quality of Life (Total)	
		Total (*n* = 831)	Nursing Home (*n* = 330)	Community (*n* = 501)	Nursing Home (*n* = 330) *M* ± *SD*; *Mdn*(IQR)	Community (*n* = 501) *M* ± *SD*; *Mdn*(IQR)	Nursing Home (*n* = 330) *M* ± *SD*; *Mdn*(IQR)	Community (*n* = 501) *M* ± *SD*; *Mdn*(IQR)	Nursing Home (*n* = 330) *M* ± *SD*; *Mdn*(IQR)	Community (*n* = 501) *M* ± *SD*; *Mdn*(IQR)
Gender %(*n*)		—	—	—	*U* = 11,265.5, *p* = 0.831	*U* = 21,255.5, *p* < 0.001	*U* = 8331.0, *p* < 0.001	*U* = 24,499.5, *p* = 0.005	*U* = 9659.0; *p* = 0.020	*U* = 28,792.0; *p* = 0.985
Male		33.5(278)	30(99)	35.7(179)	9.91 ± 3.23; 10.0(7–13)	5.93 ± 2.39; 5.0(4–8)	9.91 ± 3.23; 4.0(3–6)	3.83 ± 3.08; 3.0(2–5)	3.35 ± 0.94; 3.0(3–4)	3.52 ± 0.93; 4.0(3–4)
Female		66.5(553)	70(231)	64.3(322)	10.04 ± 3.80; 10.0(7–12)	7.34 ± 3.26; 7.0(5–10)	10.04 ± 3.63; 6.0(4–8)	4.59 ± 3.15; 4.0(2–7)	3.04 ± 1.09; 3.0(2–4)	3.50 ± 0.96; 4.0(3–4)
Age (Y; *M* ± *SD*; R)		76.5 ± 1.5 (66–99)	79 ± 9.04 (67–99)	74.9 ± 9.48 (65–97)	*r_s_* = −0.085; *p* = 0.124	*r_s_* = 0.227; *p* < 0.001	*r_s_* = 0.310; *p* < 0.001	*r_s_* = 0.250; *p* < 0.001	*r_s_* = 0.054; *p* = 0.324	*r_s_* = 0.067; *p* = 0.137
Relationship Status %(*n*)		—	—	—	χ^2^(4) = 27.123; *p* < 0.001	χ^2^(4) = 75.123; *p* < 0.001	*χ^2^*(4) = 16.038; *p* = 0.003	*χ^2^*(4) = 69.166; *p* < 0.001	*χ^2^*(4) = 11.606; *p* = 0.021	*χ^2^*(4) = 25.534; *p* < 0.001
Single		10(83)	10.9(36)	9.4(47)	8.25 ± 2.77; 7.50(6.25–10)	4.87 ± 1.99; 5.0(3–5)	4.36 ± 2.61; 4.0(3–5.75)	2.98 ± 1.76; 3.0(2–3)	3.53 ± 0.81; 4.0(3–4)	3.81 ± 0.68; 4.0(3–4)
Married		29.5(245)	9.1(30)	42.9(215)	8.67 ± 2.23; 8.5(7–10.25)	6.42 ± 2.67; 6.0(5–8)	5.70 ± 2.93; 5.0(3–9)	3.67 ± 3.02; 3.0(1–5)	3.33 ± 0.88; 3.0(3–4)	3.61 ± 0.95; 4.0(3–4)
Divorced		5.9(49)	7.3(24)	5(25)	9.08 ± 3.08; 10.0(6.25–11.75)	4.52 ± 1.66; 4.0(3–6)	5.38 ± 3.24; 5.5(2–7.75)	3.04 ± 1.24; 3.0(2–4)	3.29 ± 0.91; 3.0(3–4)	3.84 ± 0.80; 4.0(3–4)
Cohabitation		4(33)	1.5(5)	5.6(28)	6 ± 2.24; 5.0(5–7.50)	7.25 ± 3.00; 6.5(5–10)	3.00 ± 1.23; 3.0(2–4)	2.86 ± 2.51; 2.0(1.25–3)	3.60 ± 1.14; 4.0(2.5–4.5)	3.39 ± 0.88; 3.5(3–4)
Widowed		50.7(421)	71.2(235)	37.1(186)	10.62 ± 3.78; 11.0(8–13)	8.06 ± 3.30; 8.0(6–10)	5.80 ± 2.60; 6.0(4–7)	5.81 ± 3.24; 5.0(3–9)	3.02 ± 1.11; 3.0(2–4)	3.28 ± 0.99; 3.0(3–4)
Education %(*n*)		—	—	—	χ^2^(2) = 29.259; *p* < 0.001	χ^2^(2) = 4.669; *p* = 0.097	*χ^2^*(2) = 0.510; *p* = 0.775	*χ^2^*(2) = 18.059; *p* < 0.001	χ^2^(2) = 6.512; *p* = 0.039	*χ^2^*(2) = 15.534; *p* < 0.001
Elementary education		36.1(300)	45.8(151)	29.7(149)	11.13 ± 3.51; 10.9(8–13)	7.91 ± 2.82; 7.1(4.50–9)	5.72 ± 2.73; 5.1(4–7)	4.89 ± 3.06; 4.1(2–7)	3.33 ± 0.873; 3.1(2–4)	3.37 ± 0.96; 4.4(3–4)
Secondary Education		54.5(453)	46.1(152)	60.1(301)	10.82 ± 3.45; 11.0(8–13)	7.05 ± 3.09; 7.0(4.50–9)	5.63 ± 2.73; 5.0(4–7)	4.85 ± 3.25; 4.0(2–7)	3.02 ± 1.06; 3.0(2–4)	3.37 ± 0.96; 4.0(3–4)
HE (Bachelor)		8.5(71)	7.9(26)	9(45)	8.69 ± 3.69; 8.0(6–11)	6.59 ± 2.86; 6.0(5–8)	5.51 ± 2.69; 5.0(3.75–8)	3.75 ± 2.84; 3.0(2–5)	3.31 ± 1.06; 3.0(3–4)	3.68 ± 0.89; 4.0(3–4)
HE (Master or Doctoral)		0.8(7)	0.3(1)	1.2(6)	8.33 ± 2.52; 8.0(6–10.25)	6.67 ± 3.69; 6.0(4–8)	5.0 ± 2.66; 5.5(2.75–6.25)	3.67 ± 3.28; 2.0(2–4)	3.39 ± 0.85; 3.0(3–4)	3.64 ± 1.06; 4.0(3–4)
No. of CD %(*n*)		—	—	—	χ^2^(3) = 0.293; *p* = 0.961	χ^2^(3) = 1.018; *p* = 0.797	*χ*^2^(3) = 12.408; *p* = 0.006	*χ*^2^(3) = 3.998; *p* = 0.262	*χ*^2^(2) = 0.193; *p* = 0.979	*χ*^2^(2) = 5.725; *p* = 0.126
None		14.7(122)	14.2(47)	15(75)	9.98 ± 3.67; 9(7–12)	6.69 ± 2.94; 6.0(5–8)	5.34 ± 2.77; 5.0(3–7)	4.36 ± 3.32; 3.0(2–6)	3.15 ± 0.98; 3.0(3–4)	3.25 ± 1.13; 4.0(3–4)
One		52.2(434)	47.9(158)	55.1(276)	10.02 ± 3.52; 10(7–12.25)	6.88 ± 3.25; 7.0(4–9)	5.14 ± 2.48; 5.0(4–6)	4.18 ± 2.98; 3.0(2–6)	3.15 ± 1.01; 3.0(3–4)	3.54 ± 0.91; 4.0(3–4)
Two to three		31.5(262)	37(122)	27.9(140)	9.98 ± 3.82; 10.0(7–13)	6.76 ± 2.66; 7.0(5–8)	6.10 ± 2.86; 6.0(4–8)	4.44 ± 3.31; 3.0(2–7)	3.11 ± 1.16; 3.0(2–4)	3.63 ± 0.87; 4.0(3–4)
Four or more		1.6(13)	0.9(3)	2(10)	10.33 ± 1.53; 10.0(9–11)	8.00 ± 3.68; 7.0(4.75–12.25)	9.0 ± 1.73; 10.0(7–10)	6.30 ± 3.37; 6.0(3.5–9.5)	3.00 ± 1.0; 3.0(2–4)	3.0 ± 1.33; 4.0(1.75–4)
Type of CD %(*n*)		—	—	—	χ^2^(2) = 1.476; *p* = 0.478	χ^2^(2) = 1.691; *p* = 0.429	*χ^2^*(2) = 1.659; *p* = 0.436	*χ^2^*(2) = 9.832; *p* = 0.007	*χ^2^*(2) = 0.063; *p* = 0.969	*χ^2^*(2) = 2.692; *p* = 0.260
Chronic malignant dis. (cancer)		13.2(110)	4.8(16)	18.8(94)	11.31 ± 4.44; 11.5(8.25–15.25)	6.78 ± 3.05; 7.0(4.75–8)	6.44 ± 2.87; 5.5(4–9)	3.81 ± 2.88; 3.0(2–5.25)	3.13 ± 1.09; 3.0(2–4)	3.63 ± 0.84; 4.0(3–4)
Non-malignant CD		65.2(542)	73.6 (243)	59.7(299)	9.92 ± 3.60; 10.0(7–13)	6.83 ± 3.12; 7.0(4–9)	5.51 ± 2.70; 5.0(4–7)	4.32 ± 3.15; 3.0(2–7)	3.14 ± 1.08; 3.0(2–4)	3.56 ± 0.92; 4.0(3–4)
Malignant and non-malignant CD		6.9(57)	7.3(24)	6.6(33)	10.0 ± 3.41; 10.0(7.25–11.75)	7.45 ± 2.77; 7.0(6–8)	5.88 ± 2.71; 6.0(3.25–8)	5.70 ± 3.08; 5.0(3–8)	3.08 ± 1.06; 3.0(2–4)	3.30 ± 1.02; 4.0(3–4)
Experienced the following events in the past year?					—	—	—	—	—	—
Death of a loved one	Y	22.5(187)	20.6(68)	23.8(119)	9.56 ± 3.31; 9.0(8–12)	7.01 ± 3.10; 7.0(5–9)	5.56 ± 2.83; 5.0(3–7)	4.55 ± 3.24; 4.0(2–6)	3.19 ± 1.08; 3.0(2–4)	3.46 ± 1.03; 4.0(3–4)
	N	77.5(644)	79(262)	76.2(382)	10.11 ± 3.71; 10.0(7–13)	6.79 ± 3.04; 7.0(4–8)	5.56 ± 2.68: 5.0(4–7)	4.25 ± 3.11; 3.0(2–6)	3.12 ± 1.05; 3.0(2–4)	3.52 ± 0.92; 4.0(3–4)
	—				*U* = 8039.0, *p* = 0.214	*U* = 21,897.0, *p* = 0.544	*U* = 8822.5, *p* = 0.902	*U* = 21,442.5; *p* = 0.347	*U* = 8501.5; *p* = 0.549	*U* = 21,627.5; *p* = 0.387
Severe disease (self)	Y	18.5(154)	17(56)	19.6(98)	9.52 ± 3.55; 10.0(7.25–11.75)	8.48 ± 3.18; 8.0(7–10)	6.79 ± 2.93; 7.0(4.25–9)	5.84 ± 3.40; 6.0(3–9)	3.07 ± 1.14; 3.0(2–4)	3.02 ± 1.01; 3.0(2.75–4)
	N	81.5(677)	83(274)	80.4(403)	10.10 ± 3.65; 10.0(7–13)	6.44 ± 2.89; 6.0(4–8)	5.31 ± 2.60; 5.0(4–7)	3.95 ± 2.96; 3.0(2–5)	3.15 ± 1.04; 3.0(2–4)	3.63 ± 0.90; 4.0(3–4)
	—				*U* = 7026.5, *p* = 0.319	*U* = 12,003.0, *p* < 0.001	*U* = 5427.5; *p* < 0.001	*U* = 13,423.0; *p* < 0.001	*U* = 7420.0; *p* = 0.687	*U* = 12,964.5; *p* < 0.001
Severe disease (loved one)	Y	15.9(132)	16.1(53)	15.8(79)	10.60 ± 4.03; 10.0(7–13)	6.70 ± 2.70; 6.0(5–8)	5.89 ± 2.66; 6.0(4–8)	3.87 ± 2.78; 3.0(2–6)	2.92 ± 1.30; 3.0(2–4)	3.66 ± 0.70; 4.0(3–4)
	N	84.1(699)	83.9(277)	84.2(422)	9.88 ± 3.55; 10.0(7–12)	6.86 ± 3.12; 7.0(5–9)	5.49 ± 2.72; 5.0(4–7)	4.41 ± 3.20; 3.0(2–7)	3.17 ± 1.0; 3.0(2–4)	3.48 ± 0.99; 4.0(3–4)
	—				*U* = 6787.0, *p* = 0.380	*U* = 16,325.5, *p* = 0.770	*U* = 6690.0; *p* = 0.303	*U* = 15,438.5; *p* = 0.294	*U* = 6668.5; *p* = 0.272	*U* = 15,312.5; *p* = 0.214
Divorce or end of intimate relationship	Y	5.8(48)	3.6(12)	7.2(36)	13.92 ± 4.50; 16.5(13–17)	6.36 ± 2.71; 6.0(4.25–9)	5.75 ± 1.82; 6.0(4–7)	4.58 ± 3.35; 3.0(2–7.75)	1.75 ± 1.14; 1.0(1–2)	3.58 ± 0.97; 4.0(3–4)
	N	94.2(783)	96.4(318)	92.8(465)	9.85 ± 3.52; 10.0(7–12)	6.88 ± 3.08; 7.0(5–8)	5.55 ± 5.74; 5.0(4–7)	4.30 ± 3.13; 3.0(2–6)	3.19 ± 1.02; 3.0(2.75–4)	3.50 ± 0.95; 4.0(3–4)
	—				*U* = 849.5, *p* < 0.001	*U* = 7752.5, *p* = 0.458	*U* = 1740.0; *p* = 0.602	*U* = 8087.5; *p* = 0.734	*U* = 684.5; *p* < 0.001	*U* = 7870.5; *p* = 0.518
Are you satisfied with your living environment?					*U* = 9355.5, *p* < 0.001	*U* = 22,121.0, *p* = 0.657	*U* = 10,133.0; *p* = 0.071	*U* = 21,639.0; *p* = 0.426	*U* = 6471.5; *p* < 0.001	*U* = 17,769.0; *p* < 0.001
Yes		77.5(611)	69.4(229)	76.2(382)	9.45 ± 3.41; 10.0(7–12)	6.75 ± 2.84; 7.0(5–8)	5.43 ± 2.90; 5.0(3–8)	4.26 ± 3.11; 3.0(2–6)	3.39 ± 0.97; 3.0(3–4)	3.61 ± 0.87; 4.0(3–4)
No		26.5(220)	30.6(101)	23.8(119)	11.26 ± 3.82; 11.0(9–14)	7.11 ± 3.67; 7.0(5–9)	5.84 ± 2.21; 6.0(4–7)	4.52 ± 3.23; 3.0(2–7)	2.54 ± 1.02; 2.0(2–3)	3.18 ± 1.13; 3.0(2–4)
Total					10 ± 3.63; 10.0(7–12.25)	6.84 ± 3.06; 7.0(5–8.50)	5.56 ± 2.71; 5.0(4–7)	4.32 ± 3.14; 3.0(2–6)	3.13 ± 1.06; 3.0(2–4)	3.51 ± 0.95; 4.0(3–4)
Mann-Whitney U test for total					*U* = 42,104.0; *p* < 0.001		*U* = 59,181; *p* < 0.001		*U* = 64,849.5; *p* < 0.001	

Note: *n*—Sample size; %—Percent of participants; *SD*—Standard deviation; *M*—Mean; *Mdn*—Median; IQR—Interquartile range; DS—Descriptive statistics; HE—Higher Education; dis.—disease; CD—Chronic disease; Y—Year.

**Table 2 ijerph-20-04937-t002:** Spearman correlations matrix of sleep quality, frailty, and quality of life between groups.

Correlations Matrix				
		Sleep quality (Total)	Frailty (Total)	Quality of life (Total)
Nursing home	Sleep quality (Total)	1	0.418 **	−0.647 **
	Frailty (Total)	0.418 **	1	−0.435 **
	Quality of life (Total)	−0.647 **	−0.435 **	1
		Sleep quality (Total)	Frailty (Total)	Quality of life (Total)
Community	Sleep quality (Total)	1	0.467 **	−0.533 **
	Frailty (Total)	0.467 **	1	−0.524 **
	Quality of life (Total)	−0.533 **	−0.524 **	1

Note: **—Correlation is significant at the 0.01 level (2-tailed).

## Data Availability

Additional data from this study are not publicly available to maintain participants’ anonymity but can be provided on request.

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
