# Peer review of "Associations between Sleep Quality, Frailty, and Quality of Life among Older Adults in Community and Nursing Home Settings"

_ijerph, 2023, doi:10.3390/ijerph20064937_

Round 1

Reviewer 1 Report (New Reviewer)

Manuscript ijerph-2193680

Title: Associations among sleep quality, frailty, and quality of life among older adults in community and nursing home settings 

Abstract: Considering that quality of life was the dependent outcome of the study, the theoretical assumption should be described focusing on understanding factors associated with worse quality of life comparing elderly people living in the community compared with nursing homes. The same should be thought of with regard to the conclusion since the results should be described in the direction of understanding that the quality of life can be affected by factors (e.g., such as worse sleep quality and frailty) in the elderly, regardless of being a resident. from the community or nursing homes.

Introduction: The introduction should be constructed pointing out how far the literature has advanced and what the study can add again. In other words, it is necessary to make it clear what the originality of the work is. Thus, considering that it is already known that poor sleep quality and chronic conditions (multimorbidity) significantly affects older adults' quality of life (dependent outcome) in community-dwelling and nursing homes. What is the novelty of this study?

Materials and methods: This is a convenience sample. How was this sample recruited? Which locations? Which region of the country? What precautions were taken to minimize the risk of bias in the interpretation of results, given that the sample was of convenience?

What does cognitive functioning appropriate to complete the survey mean? What criterion was adopted to consider cognitive functioning appropriate?

The sample size was calculated to allow the detection of clinically meaningful associations, i.e., R2≥.02 within each group, which suggested n = 387 in each setting. How was the sample calculation done? What effect size was adopted? Effect size based on the previous study?

Why quality of life (study-dependent outcome) was not assessed using a validated questionnaire for the studied population?

Discussion: The first paragraph should summarize the main findings of the study. Thus, considering that this is a cross-sectional study, the authors should be careful when talking about a significant predictor. Suggestion: “Our findings indicate that poor sleep quality and frailty are associated with the worst quality of life among older adults living in community and nursing home settings”.

Despite the authors mentioning that this study is the first to allow direct comparisons between community-dwelling older adults or residents from nursing homes. What is the importance (relevance) and what are the gaps that this study can help to elucidate?

Noteworthy, the fact that there are already previous studies pointing to a relationship between sleep quality and frailty creates an even greater challenge with regard to pointing out the novelty of the present study.

“Some other studies support our results, e.g. that frailty [44-46] and poor sleep quality [19, 47, 48] are related to the quality of life among older adults. Our results align with previous research [33, 45, 46], which shows that health-related aspects of quality of life among older adults are related to sleep impairment. Our findings also confirm that perceived good sleep quality can influence the quality of life of older adults, even if they have 299 no sleep impairment [48].”

In addition, since the dependent outcome of the study is the quality of life, all the writing of the work must point out the possible factors associated with the quality of life of the elderly among older adults living in community and nursing home settings. It is also worth highlighting (both in the introduction and in the discussion) the guiding reasons for assessing the elderly separately for separate investigations of older adults living in community and nursing home settings.

As a strength of the study, the authors mention, “Our results show differences in the association between sleep quality and the incidence of frailty in older people in nursing homes and community-dwelling”. What would these differences be since sleep quality and frailty were both associated with quality of life both in nursing homes and in community-dwelling older adults?

Author Response

Responses to Reviewer 1

Reviewer 1 Comments

Comment 1: Title: Associations among sleep quality, frailty, and quality of life among older adults in community and nursing home settings

Response 1: Thank you for the comment.

Comment 2: Abstract: Considering that quality of life was the dependent outcome of the study, the theoretical assumption should be described focusing on understanding factors associated with worse quality of life comparing elderly people living in the community compared with nursing homes. The same should be thought of with regard to the conclusion since the results should be described in the direction of understanding that the quality of life can be affected by factors (e.g., such as worse sleep quality and frailty) in the elderly, regardless of being a resident. from the community or nursing homes.

Response 2: Thank you for the comment, and we have corrected that.

Page/line: 1/19-21

The study's results indicate that the quality of life can be affected by factors (e.g., worse sleep quality and frailty) among older adults, regardless of being a resident or from the community.

Comment 3: Introduction: The introduction should be constructed pointing out how far the literature has advanced and what the study can add again. In other words, it is necessary to make it clear what the originality of the work is. Thus, considering that it is already known that poor sleep quality and chronic conditions (multimorbidity) significantly affects older adults' quality of life (dependent outcome) in community-dwelling and nursing homes. What is the novelty of this study?

Response 3: Thank you for the comment, and we have added it.

Page/line: 2/60-70

Andersen's behavioural model [21] was used for factors associated with older adults' quality of life. The model includes contextual and individual characteristics, including demographics, social, human, and material resources, and factors such as several chronic conditions. Health behaviours include physical activity and support. The decline in one or more of a person's abilities is associated with reductions in quality of life [22].

According to the literature review, only the association between sleep quality and quality of life or frailty and quality of life has been investigated so far, and this research included only community-dwelling elderly. Because until now, there has been no research that included all three variables, we investigated the association between sleep quality, frailty and quality of life among community-dwelling older adults and nursing home residents.

Comment 4: Materials and methods: This is a convenience sample. How was this sample recruited? Which locations? Which region of the country? What precautions were taken to minimize the risk of bias in the interpretation of results, given that the sample was of convenience?

Response 4: Thank you for your comment. We added the recruited sample, locations, and region in the materials and methods section. The risk of bias we have added under the study limitation.

Page/line: 2/77-85

This study used convenience sampling to invite older adults into nursing homes and community dwellings in the North-eastern part of Slovenia. The inclusion criteria for the sample were older adults (65+ years) living in nursing homes or community dwellings and cognitive functioning appropriate to complete the survey (cognitive ability was assessed so that the healthcare professional considered which participants could participate in the study. In doing so, they looked at the absence of known organic or psychiatric affecting cognitive ability). There were no exclusion criteria. The sample size was calculated (using the Cochran formula) to allow the detection of clinically meaningful associations, i.e., R2≥.02 within each group, which suggested n = 387 (e = 95%; z = 5%) in each setting.

Page/line: 3/101-107

Data were collected from August to November 2019 in Slovenian nursing homes and the local community in the North-eastern part of Slovenia based on prior ethical approval and written permission from participating community registered nurses assisted us with preventive or curative visits to include older adults living in the community. Of the 2000 paper-based questionnaires distributed, 872 were returned. After removing 41 questionnaires because of missing data, we had 831 questionnaires fully completed for further analysis (response rate was 41.5%).

Page/line: 15/312-315

Second, recall bias can occur in a cross-sectional study. Another study limitation could present the type of sampling and its effect on the results. Due to the majority share of one gender and location, we cannot generalize the results to the whole population.

Comment 5: What does cognitive functioning appropriate to complete the survey mean? What criterion was adopted to consider cognitive functioning appropriate?

Response 5: Thank you for the comment. The cognitive ability was assessed so that the healthcare professional considered which participants could be offered participation in the study. In doing so, they looked at the absence of known organic or psychiatric affecting cognitive ability. We add this explanation.

Page/line: 2/77-85

This study used convenience sampling to invite older adults into nursing homes and community dwellings in the North-eastern part of Slovenia. The inclusion criteria for the sample were older adults (65+ years) living in nursing homes or community dwellings and cognitive functioning appropriate to complete the survey (cognitive ability was assessed so that the healthcare professional considered which participants could participate in the study. In doing so, they looked at the absence of known organic or psychiatric affecting cognitive ability). There were no exclusion criteria. The sample size was calculated (using the Cochran formula) to allow the detection of clinically meaningful associations, i.e., R2≥.02 within each group, which suggested n = 387 (e = 95%; z = 5%) in each setting.

Comment 6: The sample size was calculated to allow the detection of clinically meaningful associations, i.e., R2≥.02 within each group, which suggested n = 387 in each setting. How was the sample calculation done? What effect size was adopted? Effect size based on the previous study?

Response 6: Thank you for the comment, and we have added this explanation.

Page/line: 2/83-85

There were no exclusion criteria. The sample size was calculated (using the Cochran formula) to allow the detection of clinically meaningful associations, i.e., R2≥.02 within each group, which suggested n = 387 (e = 95%; z = 5%) in each setting.

Comment 7: Why quality of life (study-dependent outcome) was not assessed using a validated questionnaire for the studied population?

Response 7: Thank you for the comment and concerns. We use self-report questions because the quality of life can be measured by asking persons' perceptive of their quality of life without a detailed split. We also use Cronbach α to that's our questionnaire.

Comment 8: Discussion: The first paragraph should summarize the main findings of the study. Thus, considering that this is a cross-sectional study, the authors should be careful when talking about a significant predictor. Suggestion: "Our findings indicate that poor sleep quality and frailty are associated with the worst quality of life among older adults living in community and nursing home settings".

Response 8: Thank you for the comment; we have corrected it.

Page/line: 10/230-233

Our findings indicate that poor sleep quality and frailty are associated with the quality of life among older adults living in community and nursing home settings. This study allows direct comparisons between community-dwelling older adults or residents from nursing homes for all studied variables.

Comment 9: Despite the authors mentioning that this study is the first to allow direct comparisons between community-dwelling older adults or residents from nursing homes. What is the importance (relevance) and what are the gaps that this study can help to elucidate?

Response 9: Thank you for the comment. It is important because it took place simultaneously in both groups, and no differences could be detected. It is, however, to establish that the selected factors influence/are connected or are significant in both groups. Still, at the same time, the results open us up for new research in more precise clarification of other factors contributing to sleep quality, fragility and quality of life.

Comment 10: Noteworthy, the fact that there are already previous studies pointing to a relationship between sleep quality and frailty creates an even greater challenge with regard to pointing out the novelty of the present study.

"Some other studies support our results, e.g. that frailty [44-46] and poor sleep quality [19, 47, 48] are related to the quality of life among older adults. Our results align with previous research [33, 45, 46], which shows that health-related quality of life among older adults is related to sleep impairment. Our findings also confirm that perceived good sleep quality can influence the quality of life of older adults, even if they have 299 no sleep impairment [48]."

Response 10: Thank you for the comment. The presented studies deal with only two studied variables. Also, they include only older people in the community, while our research includes all three studied variables on a sample of older people in the community and in an institution. We also determined the importance of behavioural model factors for quality of life.

Comment 11: In addition, since the dependent outcome of the study is the quality of life, all the writing of the work must point out the possible factors associated with the quality of life of the elderly among older adults living in community and nursing home settings. It is also worth highlighting (both in the introduction and in the discussion) the guiding reasons for assessing the elderly separately for separate investigations of older adults living in community and nursing home settings.

Response 11: Thank you for the comment, and we have added it.

Comment 12: As a strength of the study, the authors mention, "Our results show differences in the association between sleep quality and the incidence of frailty in older people in nursing homes and community-dwelling". What would these differences be since sleep quality and frailty were both associated with quality of life both in nursing homes and in community-dwelling older adults?

Response 12: Thank you for the comment. Our results show differences in sleep quality, the incidence of frailty and the quality of life in older people in nursing homes and community-dwelling, which will help identify potential factors related to the quality of life and provide us with guidelines for future research.

Comment about English language and style: With the assistance of a senior researcher, a native speaker of English, we have extensively rewritten our manuscript for clarity and accuracy.

Reviewer 2 Report (Previous Reviewer 1)

A similar problem has been raised in the first review comments. The author did not give effective modifications. How did the author get the results on page 8, lines 237-239? There is no corresponding content displayed in the table. Is it the result of linear regression with sleep quality as dependent variable and only sleep quality and frailty as independent variables? This may not be appropriate. Because adjustment factors and the interaction between sleep quality and weakness were not considered. In addition, Table 1 and page 8, lines 161 and 162, "c2(2)" should be replaced by "c2(2)".

Author Response

Responses to Reviewer 2

Reviewer 2 Comments

Comment 1: A similar problem has been raised in the first review comments. The author did not give effective modifications. How did the author get the results on page 8, lines 237-239? There is no corresponding content displayed in the table. Is it the result of linear regression with sleep quality as dependent variable and only sleep quality and frailty as independent variables? This may not be appropriate. Because adjustment factors and the interaction between sleep quality and weakness were not considered. In addition, Table 1 and page 8, lines 161 and 162, "c2(2)" should be replaced by "c2(2)".

Response 1: Thank you for your comment. We agree with you, and for that reason we have we have removed the linear regression from the article.

Comment about English language and style: With the assistance of a senior researcher, a native speaker of English, we have extensively rewritten our manuscript for clarity and accuracy.

Reviewer 3 Report (New Reviewer)

The article examines the relationship between sleep quality, frailty, and quality of life among older adults who are community dwellers and nursing home residents in Slovenia. An exhaustive data analysis has been done to examine the influence of frailty on older people's quality of life in nursing homes and community settings, as well as the relationship between sleep quality and its occurrence. The findings showed a substantial relationship between inadequate sleep and frailty as well as a reduction in quality of life. However, the results among the people living in nursing homes and community residences were found to be noticeably different. Overall, this paper seems nice.

The article shows a rigorous study. The introduction, methodology, results, and discussion were presented in detail. Also, the results are discussed nicely in light of relevant literature.

The study region is stated as Slovenian nursing homes and local communities. Line 320 states some limitations regarding the key population characteristics. A clarification for the following is expected:

  1. From which part (or region) of Slovenia has the survey data been collected? Are all regions of Slovenia equally represented by the data?

  2. What is the possibility of generalizability of the study results to other countries/regions?

Author Response

Responses to Reviewer 3

Reviewer 3 Comments

Comment 1: The article examines the relationship between sleep quality, frailty, and quality of life among older adults who are community dwellers and nursing home residents in Slovenia. An exhaustive data analysis has been done to examine the influence of frailty on older people's quality of life in nursing homes and community settings, as well as the relationship between sleep quality and its occurrence. The findings showed a substantial relationship between inadequate sleep and frailty as well as a reduction in quality of life. However, the results among the people living in nursing homes and community residences were found to be noticeably different. Overall, this paper seems nice.

Response 1: Thank you for the comment.

Comment 2: The article shows a rigorous study. The introduction, methodology, results, and discussion were presented in detail. Also, the results are discussed nicely in light of relevant literature.

Response 2: Thank you for the comment.

Comment 3: The study region is stated as Slovenian nursing homes and local communities. Line 320 states some limitations regarding the key population characteristics. A clarification for the following is expected:

  • Comment 3.1: From which part (or region) of Slovenia has the survey data been collected? Are all regions of Slovenia equally represented by the data?
  • Response 3.1: Thank you for your comment. The materials and methods section added the recruited sample, locations, and region.

Page/line: 2/77-85

This study used convenience sampling to invite older adults into nursing homes and community dwellings in the North-eastern part of Slovenia. The inclusion criteria for the sample were older adults (65+ years) living in nursing homes or community dwellings and cognitive functioning appropriate to complete the survey (cognitive ability was assessed so that the healthcare professional considered which participants could participate in the study. In doing so, they looked at the absence of known organic or psychiatric affecting cognitive ability). There were no exclusion criteria. The sample size was calculated (using the Cochran formula) to allow the detection of clinically meaningful associations, i.e., R2≥.02 within each group, which suggested n = 387 (e = 95%; z = 5%) in each setting.

Page/line: 3/101-107

Data were collected from August to November 2019 in Slovenian nursing homes and the local community in the North-eastern part of Slovenia based on prior ethical approval and written permission from participating community registered nurses assisted us with preventive or curative visits to include older adults living in the community. Of the 2000 paper-based questionnaires distributed, 872 were returned. After removing 41 questionnaires because of missing data, we had 831 questionnaires fully completed for further analysis (response rate was 41.5%).

  • Comment 3.2: What is the possibility of generalizability of the study results to other countries/regions?
  • Response 3.1: Thank you for your comment. In the materials and methods section, we have addressed this concern under limitations.

Page/line: 15/312-315

Second, recall bias can occur in a cross-sectional study. Another study limitation could present the type of sampling and its effect on the results. Due to the majority share of one gender and location, we cannot generalize the results to the whole population.

Comment about English language and style: With the assistance of a senior researcher, a native speaker of English, we have extensively rewritten our manuscript for clarity and accuracy.

Round 2

Reviewer 1 Report (New Reviewer)

The aurhors answered point-by-point. Overall recmendation: accept in the present form.

This manuscript is a resubmission of an earlier submission. The following is a list of the peer review reports and author responses from that submission.

Round 1

Reviewer 1 Report

This manuscript focuses on the correlation between sleep quality, frailty and quality of life among older adults in community and nursing home. These three aspects are worthy of attention in the study of older adults. However, this manuscript needs better research design and statistical analysis. (1) “The sample size was calculated using the Cochran formula. Based on that, we calculated the approximately the representative sample should be n = 384 (e = 95%; z = 5%) (Page 2, Line75-77)”. Is 384 a sample size for all older adults or a sample size for the number of older adults in the community or nursing homes? What do e and z mean? Is there any reference for the calculation method of sample size? (2) In the Materials and Methods section, the quality of life is not described in detail. How is the quality of life measured? How many self-reported problems does quality of life include? If there is only 1 problem, how to calculate Cronbach α coefficient  (Page 3, Line113)? In addition, the measurement methods of other variables should also be described, such as physical activity. (3) “Linear logistic regression analyses were used… (Page 3, Line108)”. Please pay attention to whether it is linear regression? The author should also pay attention to whether the data of this manuscript are suitable for linear regression analysis? (4) In Table 1, the total number of variables is not 831, such as Education, Type of CD. (5) The focus of this manuscript is the correlation between sleep quality, frailty and quality of life. However, this content is less involved in the results. In addition, the influence of confounding factors and the interaction between sleep quality and weakness were not considered. (6) In the linear regression model, with more variables introduced, the R2 for the model would undoubtedly increase. And the R2 can only indicate the percentage of variables explaining for the model. (7) What is the highlight of this article? The results are not novel enough.  

Author Response

Reviewer 1 Comments

Methods

Comment 1: The sample size was calculated using the Cochran formula. Based on that, we calculated the approximately the representative sample should be n = 384 (e = 95%; z = 5%) (Page 2, Line75-77). Is 384 a sample size for all older adults or a sample size for the number of older adults in the community or nursing homes? What do e and z mean? Is there any reference for the calculation method of sample size?

Response 1: The total representative sample was calculated based on older people living in Slovenia, and we did not separate them in particular during the calculation. e stands for Confidence Level (%), and z for Margin of Error (%). A reference to the Cochran formula is added in the manuscript.

Comment 2: In the Materials and Methods section, the quality of life is not described in detail. (1) How is the quality of life measured? (2) How many self-reported problems does quality of life include? (3) If there is only 1 problem, how to calculate Cronbach α coefficient (Page 3, Line113)? In addition, the measurement methods of other variables should also be described, such as physical activity.

Response 2: (1) The quality of life question is measured on a self-assessment Likert scale. (2) We did not use a validated questionnaire to measure the quality of life. Therefore, we did not determine self-report problems. (3) Cronbach α coefficient was calculated based on two validated questionnaires. (4) The variable physical activities have been added to the method section ("In the study, a combination of the Pittsburgh Sleep Quality Index (PSQI) [22], the Til-burg Frailty Indicator (TFI) [23], the Quality of life self-assess question on a five-point Lik-ert scale (from 1 "very poor" to 5 "very good") and an open-ended question about Physical activity in the last seven days were used.”).

Comment 3: "Linear logistic regression analyses were used… (Page 3, Line108)". Please pay attention to whether it is linear regression? The author should also pay attention to whether the data of this manuscript are suitable for linear regression analysis?

Response 3: Thank you for your comment. We have corrected the grammatical error. We have taken into consideration the appropriateness of using regression.

Comment 4: In Table 1, the total number of variables is not 831, such as Education, Type of CD.

Response 4: There was a grammatical error when entering the data into the table. We have now added the missing line to the table. The total of variables the Type of CD does not match because only those who have indicated a chronic disease are listed.

Comment 5: The focus of this manuscript is the correlation between sleep quality, frailty and quality of life. However, this content is less involved in the results. In addition, the influence of confounding factors and the interaction between sleep quality and weakness were not considered.

Response 5: Thank you for your comment. We agree that we have not taken the impact into account, which we have now added under the limitations of the survey.

Comment 6: In the linear regression model, with more variables introduced, the R2 for the model would undoubtedly increase. And the R2 can only indicate the percentage of variables explaining for the model.

Response 6: Thank you for your comment, and we have taken note.

Comment 7: What is the highlight of this article? The results are not novel enough. 

Response 7: We found that poor sleep quality is associated with frailty and quality of life among community-dwelling older people and nursing home residents. Therefore, if the quality of sleep is improved, the elderly will be less fragile, and, as a result, their quality of life will be improved. Furthermore, our results show differences in the association between sleep quality and the incidence of frailty in older people in nursing homes and community-dwelling. This helps identify potential factors related to the quality of life and provides guidelines for future research.

Author Response

Responses to Reviewer 2

Reviewer 2

Comment 1: Do the authors have a list of medications used by the older adults?

Response 1:Thank you for your comment. We have not collected this information as this was not the study's primary aim. We have collected the data requested from the instrument used. We agree that collating this information can add new value to this research topic and open new opportunities for further research.

Comment 2: We know that many older adults may have sleep disorders, how did the authors correlate or isolate this subpopulation? with these older adults who did not use medication for sleep disorders (insomnia for example)?

Response 2: Thank you for the comment. We have isolated sleep quality and sleep disorders with the help of the Pittsburgh Sleep Quality Index, which overall scores range from 0 to 21, indicating poorer sleep quality and a high level of sleep disorders. A global score ≥5 indicates clinically significant sleep impairment. In the results section, we added data about needed medication for sleep disorders by older people: "Among 831 older people, 551 did not need medication for sleep disorders. One hundred fifty older people need medication for insomnia regularly, and 130 older people less than once a week".

Comment 3: The authors cite frailty, but where is the frailty assessment or scores and what is the method for using frailty?

Response 3: Thank you for your comment. Those data are presented in the method section ("TFI is a 15-item questionnaire measuring frailty among older adults. TFI consists of three components (physical, psychological and social components). Total scores range from 0 to 15 [23], and s total score ≥5 points indicates frailty [23, 24].").

Comment 4: The authors describe chronic diseases, number and type. I ask which chronic diseases are most found in your population?

Response 4: Thank you for your comment. We have added the type of chronic disease in the results section.

Comment 5: What criteria did the authors use to assess physical activity, any questionnaire?

Response 5: Thank you for your comment. The questionnaire asked older people to self-assess their physical activity in the last seven days. We have added a description of this issue in the methods section.

Comment 6: Be consistent in frailty/fragility terms used, please.

Response 6: Thank you for your comment. The term has been corrected and standardised.

Round 2

Reviewer 1 Report

I am generally satisfied with the revision of the article. I have no other review comments.

Author Response

Thank you for your comment and review. 

Reviewer 2 Report

Upgrading the manuscript was not enough.

Author Response

Thank you for your comment. We have improved the manuscript throughout the whole document. 
